# FACE-FEATURE TUNING: POST-PROCESSING CALIBRATION FOR FAIR AND ACCURATE DEEPFAKE DETECTION

## ABSTRACT

Deepfake detectors show large performance gaps across demographic groups. Existing fairness approaches require demographic labels, retraining, or sacrifice accuracy. We introduce Face-Fairness (FF), a plug-and-play framework for bias mitigation. Our primary contribution, Face-Feature Tuning (FFT), is the first demographic label-free fairness method demonstrated for deepfake detection: a lightweight calibrator that performs a logit remapping conditioned on frozen face embeddings, trained on a held-out validation split. We complement FFT with two thresholding variants: FF-Max, which maximizes worst-group accuracy when demographics are available, and FF-Discover, which does the same with embedding-discovered groups. Across in-domain and cross-dataset test settings, FF consistently reduces FPR/TPR gaps and improves minimum group accuracy while maintaining (often improving) overall accuracy. The approach is detector-agnostic, adds negligible runtime overhead, and requires no access to identity attributes.

## 1 INTRODUCTION

The proliferation of deepfakes has created an arms race between generation and detection methods. While modern detectors achieve reasonable aggregate performance, recent work reveals detection accuracy may vary by over 40% across demographic groups (Trinh & Liu, 2021; Hazirbas et al., 2021; Xu et al., 2024). Trinh & Liu (2021) report that one detector's false-positive rate was nearly twice as high for female vs. male subjects (14.0% vs. 7.7%), and African subgroups—especially males—exhibits the lowest true-positive rates by 4.7–10.7 percentage points, implying higher false-accusation risk for some groups and less protection against deepfakes for others.

These fairness gaps pose deployment challenges. Content moderation systems may disproportionately flag authentic videos from minority groups as fake (Trinh & Liu, 2021; Hazirbas et al., 2021). Identity verification would be more likely to fail for certain demographics (Klare et al., 2012; Buolamwini & Gebru, 2018). Adversaries can exploit these blind spots by targeting vulnerable populations with detection-evading deepfakes (Xu et al., 2018; Yan et al., 2024; Chesney & Citron, 2019). As deepfake detectors become embedded in infrastructure, from social media platforms to legal evidence systems, addressing these biases is a practical necessity (Dolhansky et al., 2020; Chesney & Citron, 2019).

Existing pre-processing methods that balance training data may discard valuable samples and require significant manual effort and compute (Kamiran & Calders, 2012; Idrissi et al., 2022; Ju et al., 2023). In-processing methods that modify training objectives typically require model retraining and, in many cases, explicit demographic annotations (Ju et al., 2023; Lin et al., 2024; Liu et al., 2021). Post-processing methods that adjust decision thresholds often need group labels at inference time and can reduce overall accuracy under fairness constraints (Hardt et al., 2016; Chen & Wu, 2020; Menon & Williamson, 2017; Kleinberg et al., 2016). Further, methods tuned for one dataset or manipulation family can harm fairness on unseen generators Lin et al. (2024). These limitations make current solutions impractical for organizations using commercial base models.

Our primary contribution, Face-Feature Tuning (FFT), offers a novel solution to the deployment challenge: it achieves fairness without demographic labels, without retraining, and without sacrificing accuracy. FFT's key insight is that face embeddings from pre-trained models (e.g., ArcFace (Deng et al., 2022)) implicitly encode the visual patterns that correlate with detector failures in a continuous representation. By training a lightweight neural network to learn better decision boundaries based on

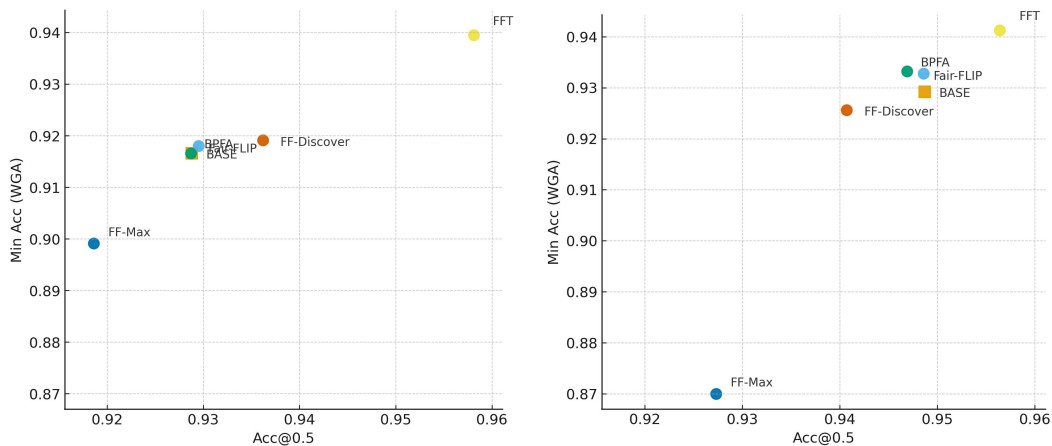

Figure 1: OpenForensics fairness–utility comparison (Le et al., 2021). **Left:** MobileNetV3-Small. **Right:** Xception. BASE (square); others (circles): FFT (ours), Fair-FLIP (Szandala et al., 2025), BPFA (Liu et al., 2025), FF-Max, FF-Discover. X-axis: accuracy@0.5; Y-axis: worst-group accuracy (14 gender/ethnicity groups). FFT achieves best overall and worst-group accuracy on both backbones.

embeddings, FFT recalibrates predictions where systematic biases occur while preserving reliable decisions elsewhere. This can dramatically improve performance across groups as seen in Figure 1.

While FFT consistently provides the highest fairness and accuracy, our framework also includes FF-Max and FF-Discover as simpler alternatives. These methods serve as strong baselines and are highly practical in deployment environments where training or running even a lightweight neural network is a barrier. FF-Max directly optimizes thresholds using known group labels, while FF-Discover does so over groups found via clustering the embedding space. This makes them easy to implement and as we will discuss in more detail potentially more robust under extreme dataset shifts.

Table 1 positions our FF framework within the landscape of existing methods. While recent approaches have made important contributions—DAG-FDD introduced demographic-agnostic training objectives (Ju et al., 2023), DAID established causal links between fairness and generalization (Cheng et al., 2025), and Fair-FLIP showed promise in post-processing feature adjustment (Szandala et al., 2025)—each faces limitations that restrict practical deployment given their dependence on either retraining or demographic labels. Our evaluation demonstrates the effectiveness of FF. FFT reduces performance gaps across groups while improving the original detector's accuracy. Compared to baseline methods, FF consistently achieves state-of-the-art results in both fairness and overall accuracy across all evaluation metrics.

The contributions of this work are: **FF-Max**, a post-processing method that learns *group-specific* decision thresholds to maximize worst-group accuracy when demographic labels are available at calibration and inference, requiring no retraining of the base detector; **FF-Discover**, a post-processing method that discovers *latent groups* by clustering face embeddings and then learns cluster-conditional thresholds to maximize minimum group accuracy without demographic labels at any stage and without retraining; and **FFT**, a post-processing calibration method that trains a lightweight MLP on top of frozen face embeddings and the base detector logit to learn a better decision boundary, which is group-label-free, retraining-free, and improves both worst-group and overall accuracy.

## 2 RELATED WORK

### 2.1 FAIRNESS AND GENERALIZATION IN DEEPFAKE DETECTION

Early work showed that modern deepfake detectors exhibit demographic disparities even when overall accuracy is high. Trinh & Liu (2021) systematically evaluated three popular detectors and reported sizable error-rate gaps across racial groups, while also noting that training datasets are overwhelmingly composed of Caucasian subjects—conditions that can induce spurious correlations

Table 1: Comparison of methods.

| Method | Category | Group-Label-Free? | Retraining-Free? | Key Mechanism |
|---|---|---|---|---|
| *Existing Methods* | | | | |
| DAG-FDD (Ju et al., 2023) | In-processing | ✓ | ✗ | CVaR on latent groups |
| DAW-FDD (Ju et al., 2023) | In-processing | ✗ | ✗ | Hierarchical CVaR |
| DAID (Cheng et al., 2025) | In-processing | ✗ | ✗ | Group reweighting |
| PG-FDD (Lin et al., 2024) | In-processing | ✗ | ✗ | Fairness Loss |
| BPFA (Liu et al., 2025) | Post-processing | ✗ | ✓ | Conv pruning |
| Fair-FLIP (Szandala et al., 2025) | Post-processing | ✗ | ✓ | Reweighting |
| *Face-Fairness (FF) Framework (Ours)* | | | | |
| **FF-Max** | Post-processing | ✗ | ✓ | Group thresholding |
| **FF-Discover** | Post-processing | ✓ | ✓ | Group thresholding |
| **FFT** | Post-processing | ✓ | ✓ | Face Feature Tuning |

between demographic cues and "fakeness". Hazirbas et al. (2021) analyzed top detector systems and likewise found lower performance on darker skin tones. Subsequent large-scale annotation efforts (e.g., demographic and non-demographic attributes across popular datasets) confirmed substantial dataset skew and detector sensitivity to attributes unrelated to manipulation Xu et al. (2024).

Detectors often fail to generalize fairly across datasets and manipulation families. Cross-dataset studies routinely observe sharp AUROC drops when training on one generator family and testing on another Nadimpalli & Rattani (2022); Lin et al. (2024). This motivates methods that explicitly address both fairness and generalization rather than optimizing either in isolation Ju et al. (2023); Ma et al. (2025).

## 2.2 MITIGATION STRATEGIES

### 2.2.1 PRE-PROCESSING

These methods act on the data before training—e.g., reweighting or resampling to balance groups, or augmenting with synthetics—without changing the detector architecture or loss.

Two directions are common: (i) curating or annotating balanced datasets and (ii) synthesizing data to fill demographic gaps. Gender-Balanced DeepFake (GBDF) introduced gender labels and class balancing to reduce performance gaps, albeit with limited improvements when used alone (Nadimpalli & Rattani, 2022). Mass annotation work further enabled systematic fairness evaluation across dozens of attributes (Xu et al., 2024). Synthetic face generation methods (e.g., controllable GAN+diffusion pipelines) allow fine control over demographic proportions and have been proposed to support fairness-aware training; however, synthetic faces cannot substitute for real identities.

Empirically, synthetic-only training exhibits a persistent domain gap and reduced intra-/inter-class diversity relative to real datasets, with lower accuracy and unstable transfer (Qiu et al., 2021; Boutros et al., 2023). Detectors can overfit to generator-specific artifacts, and fairness gains from synthetic augmentations may fail to carry over to real footage (Kamat et al., 2023; Amerini et al., 2025).

### 2.2.2 IN-PROCESSING

These methods modify the training objective and/or optimization to encourage fairness.

Let $\theta$ be detector parameters and $\ell(\theta; X, Y)$ a per-example loss for input $X$ and label $Y \in \{0, 1\}$. We write $(t)_+ = \max\{t, 0\}$, and $\alpha \in (0, 1]$ for a tail level. When demographic labels are used, $D_i$ is the group of sample $i$. We denote demographic features by $d$, domain-agnostic forgery features by $f^g$, and domain-specific forgery features by $f^a$.

**DAG–FDD & DAW–FDD:** Ju et al. (2023) propose two CVaR-based training losses that plug into standard deepfake detectors. The demographic-agnostic variant DAG–FDD minimizes an empirical CVaR risk $\mathrm{CVaR}_\alpha(\theta) = \inf_{\lambda \in \mathbb{R}} \{\lambda + \frac{1}{\alpha}\mathbb{E}[(\ell(\theta; X, Y) - \lambda)_+]\}$, which upper-bounds the worst-case latent-group risk and protecting any subgroup with a prevalence $> \alpha$ without group labels. The demographic-aware variant DAW–FDD nests CVaR: an outer group-level encourages parity across specified groups, while an inner per-group CVaR handles real/fake imbalances within each group.

**PG–FDD:** Lin et al. (2024) target fairness generalization by learning disentangled features—demographic $d$, domain-agnostic forgery $f^g$, and domain-specific forgery $f^a$—and then training with a distribution-aware fairness loss plus sharpness-aware optimization. The demographic head uses a label-distribution-aware margin. To encourage a larger margin for groups containing fewer examples, alongside the use of SAM to flatten sharp minima.

**DAID:** Cheng et al. (2025) present a causal framing that treats "fairness" as a treatment and "generalization" as the outcome; using back-door adjustment they write the target as $P(Y \mid \mathrm{do}(\text{fairness})) = \sum_c P(Y \mid \text{fairness}, c) P(c)$, where $c$ collects confounders such as demographic distribution and model capacity. The proposed DAID implementation is intended to realize this via two in-processing modules: (i) demographic-aware data rebalancing and (ii) demographic-agnostic feature aggregation that aligns same-label features across different groups with a cosine-similarity loss applied in a low-rank projected subspace (a trainable projection acts as a filter; a regularizer prevents collapse).

### 2.2.3 Post-processing

When retraining or training a large detector is impractical, post–processing can reduce disparities by acting on a trained model's outputs or its last–layer interface, e.g., reweighting final–layer inputs or pruning biased activations, while leaving the backbone untouched.

**Fair–FLIP:** Szandala et al. (2025) proposed a model–agnostic, post–hoc reweighting of a detector's final–layer inputs that downweights features with high variance across demographic subgroups and upweights stable ones. They estimate between–group variability for each penultimate–layer activation $f_i$ on a small, demographically annotated set, then modify the corresponding classifier weight via $w_i' = w_i \cdot (1 + \alpha - \hat{\sigma}(f_i))$, where $\hat{\sigma}(f_i)$ is a normalized across–group dispersion and $\alpha$ controls the strength of prioritization. This requires no retraining and no demographic labels.

**Pruning/activation–based adjustments (BPFA).** Liu et al. (2025) introduce the FairFD evaluation suite and a post–processing method, *Bias Pruning with Fair Activations* (BPFA), which aims to rebalance groupwise errors by selecting "fair" activations and suppressing biased ones; the authors emphasize that it works without retraining or weight updates and report improved fairness when applied to existing detectors. Related work characterizes BPFA as pruning biased network components based on activation statistics—an inference–time adjustment that can trade a small amount of utility for reduced disparities—underscoring its role as a plug–in, backbone–preserving step.

### 2.3 Positioning of Our Contributions.

Training-time approaches (e.g., DAG-FDD/DAW-FDD/DAID) can improve fairness but require full retraining, explicit demographic labels, and access to model internals—requirements that conflict with production release cycles and compute budgets. Post-processing baselines often still assume group labels at validation time or rely on architecture-level hooks which are frequently unavailable.

FFT calibrates the outputs of frozen deepfake detectors using off-the-shelf face embeddings. Because these embeddings encode demographic and other relevant structure, our calibrator implicitly infers protected-attribute proxies and apply sample-specific adjustments that correct systematic group errors.

When group labels are available, FF-Max extends worst-case optimization to detector calibration to mitigate error for the most disadvantaged group. Both it and its label-free counterpart, FF-Discover, are computationally lightweight, relying on group thresholding. Our approaches enable a fast turn-around as attacks and datasets evolve. Updating to new attacks involves refitting a small module on recent data, without full retraining or architecture changes, or demographic annotation.

## 3 METHODS

### 3.1 PROBLEM FORMULATION

Let $\mathcal{D} = \{(x_i, y_i)\}_{i=1}^n$ denote a dataset of face images $x_i \in \mathcal{X}$ with labels $y_i \in \{0, 1\}$ (real=0, fake=1), where each image belongs to a demographic group $g_i \in \mathcal{G}$ (either observed or latent). A deepfake detector $f_\theta : \mathcal{X} \to [0, 1]$ outputs scores interpreted as fake probabilities. We quantify fairness violations across demographic groups through disparity metrics: $\Delta_{\text{FPR}} = \max_{g,g'} |\text{FPR}_g - \text{FPR}_{g'}|$ and $\Delta_{\text{TPR}} = \max_{g,g'} |\text{TPR}_g - \text{TPR}_{g'}|$, where FPR and TPR denote false positive and true positive rates respectively. Additionally, we track the worst-group accuracy $\min_g \text{Acc}_g$ to ensure no demographic group experiences disproportionately poor performance, and other fairness metrics.

### 3.2 FACE-FEATURE TUNING

Let $Z(x) = \text{logit}(f_\theta(x))$ denote the base detector's logit output and $\phi(x) \in \mathbb{R}^d$ the frozen ArcFace embedding with $d = 512$. FFT learns a shallow multi-layer perceptron $g_\psi : \mathbb{R}^{d+1} \to \mathbb{R}$ that maps the concatenated embedding and base logit to the final prediction: $\tilde{Z}(x) = g_\psi([\phi(x), Z(x)])$. The calibrated probability is then $\hat{p}(x) = \sigma(\tilde{Z}(x))$, where $\sigma$ denotes the sigmoid function.

The prediction head $g_\psi$ uses a two-layer architecture with hidden dimensions $128 \to 64$, ReLU activations, and a dropout rate of $0.15$ to prevent overfitting. Both the base detector $f_\theta$ and face encoder $\phi$ remain frozen. We train $g_\psi$ on a held-out validation split using binary cross-entropy: $\mathcal{L}_{\text{BCE}} = -\frac{1}{n} \sum_{i=1}^n [y_i \log \sigma(\tilde{Z}_i) + (1 - y_i) \log(1 - \sigma(\tilde{Z}_i))]$, where $\tilde{Z}_i = g_\psi([\phi(x_i), Z(x_i)])$ is the logit output. Validation is partitioned $90/10$ for hyperparameter selection and early stopping. Empirically, this two-layer head performed moderately better than logistic regression, and deeper MLP variants did not provide a consistent benefit.

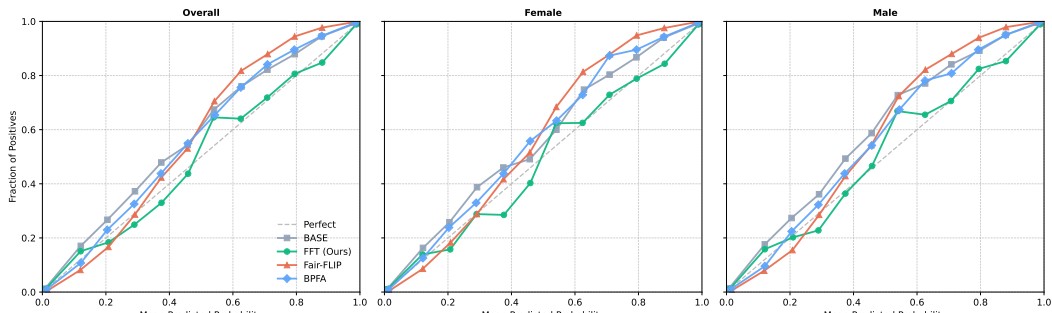

Figure 2: Calibration curves on the *OpenForensics* dataset (Le et al., 2021) on Xception base model. FFT achieves substantial ECE reduction compared to the baseline (73% overall, 65% for females, 76% for males), while existing post-processing methods Fair-FLIP and BPFA degrade calibration.

#### 3.2.1 INTUITION

FFT is a conditional calibrator, that instead of learning a univariate map $Z(x) \mapsto \hat{p}(x)$ which assumes $p(y=1 \mid Z)$ is invariant across subpopulations, it learns $p(y=1 \mid Z, \phi)$ by applying a residual correction $\tilde{Z}(x) = Z(x) + r_\psi(\phi(x), Z(x))$. This view makes FFT a continuous label-free generalization of group-conditioned thresholds: with constant $(a_k, b_k, c_k)$ it reduces to global Platt scaling; when $a_k \approx 1$ and $b_k$ is small it behaves like per-group offsets; when $a_k$ varies it attenuates or amplifies over/under-confident base scores.

Here, FF-Max and FF-Discover can be seen as hard-coded special cases, while FFT learns the encoding and corrections jointly and implicitly. The effect is visible in Figure 2: FFT tracks the diagonal more closely than the base detector and decreases calibration error significantly. By delivering better-conditioned probabilities, a single threshold yields fewer systematic false positives/negatives in specific embedding neighborhoods, shrinking subgroup gaps while preserving the global ranking signal from $Z(x)$—which explains the simultaneous accuracy and fairness gains in our experiments.

### 3.3 Face-Feature-Max: Group-Conditional Threshold Optimization

When demographic labels are available during calibration and inference, FF-Max learns group-specific decision thresholds that maximize worst-group accuracy. Define the decision rule as $\hat{y}(x; \mathbf{t}) = \mathbb{1}\{Z(x) \geq t_{g(x)}\}$, where $\mathbf{t} = \{t_g\}_{g \in \mathcal{G}} \in [0,1]^{|\mathcal{G}|}$ represents per-group thresholds. On the validation set, we solve the max-min optimization problem: $\max_{\mathbf{t} \in [0,1]^{|\mathcal{G}|}} \min_{g \in \mathcal{G}} \mathrm{Acc}_g(\hat{y}(\cdot; \mathbf{t}))$. In practice, we enumerate candidate thresholds on a grid (step size 0.01) for each group, compute the Pareto frontier of accuracy versus minimum group accuracy, and select the threshold vector $\mathbf{t}^*$ that maximizes $\min_g \mathrm{Acc}_g$. Learned thresholds are fixed and applied at test time based on known groups.

### 3.4 Face-Feature-Discover: Clustering-Based Fair Thresholding

When demographic annotations are unavailable, FF-Discover automatically discovers latent groups through unsupervised clustering of face embeddings. We apply $K$-means clustering on $L_2$-normalized ArcFace features: $c(x) = \arg\min_{k \in \{1, \dots, K\}} \|\phi(x) - \mu_k\|_2$, where cluster centroids $\{\mu_k\}$ minimize $\sum_k \sum_{x_i \in C_k} \|\phi(x_i) - \mu_k\|_2^2$.

We select $K$ through silhouette analysis over the range $K \in \{6, 8, 10\}$. After clustering, we learn cluster-specific thresholds $\mathbf{t} = \{t_k\}_{k=1}^K$ by maximizing $\min_k \mathrm{Acc}_k$ on validation data, following the same optimization procedure as FF-Max but with discovered clusters replacing demographic groups. At test time, we assign each sample to its nearest centroid and apply the corresponding threshold. Conceptually, this is similar to FairCal which used for face verification rather than deepfake detection. FairCal forms latent subgroups by $K$-means over face embeddings and performs per-cluster calibration for face verification rather than deepfake detection Salvador et al. (2022).

### 3.5 Implementation Details

**Base Detectors.** We evaluate two architectures: Xception (Chollet, 2017), a depth-wise separable CNN that remains a common baseline in deepfake detection and MobileNetV3-Small (Howard et al., 2019), used to stress-test our method in a low-capacity settings. MobileNetV3 was explicitly designed via hardware-aware NAS/NetAdapt for on-device, low-resource use with low mobile-CPU latency, and recent deepfake-detection work increasingly emphasizes real-time/edge settings where lightweight backbones are required (Romeo et al., 2024).

We follow standard two-stage fine-tuning: head-only warmup (2 epochs) followed by full model fine-tuning with AdamW optimizer, cosine learning rate schedule with linear warmup, gradient clipping (max norm 1.0), and exponential moving average of weights (decay 0.999). The FFT calibrator $g_\psi$ is trained for 20 epochs with AdamW (lr $= 2 \times 10^{-3}$, weight decay $5 \times 10^{-4}$). Training typically converges within 10-15 epochs. We selected the best model by AUC on the validation data.

## 4 Experiments

We evaluate on two complementary benchmarks: OpenForensics and FaceForensics++ (FF++). **OpenForensics** is a large-scale, in-the-wild, multi-face forgery dataset with face-wise annotations designed for detection and segmentation under real-world clutter and multi-person scenes (Le et al., 2021). Following Szandala et al. (2025): keep the Kaggle OpenForensics train/validation/test partition, run detectors at the face-instance level, and use the validation split to fit post-processing parameters. **FF++** is a benchmark featuring canonical manipulation families—DeepFakes (DF), FaceSwap (FS),FaceShifter (FST), Face2Face (F2F), and NeuralTextures (NT)—with standardized compression settings (c0/c23/c40) (Rössler et al., 2019). We use the official splits and report video-level metrics obtained by averaging frame probabilities per video we use c23 as this compression level is a standard benchmark setting that realistically simulates the video quality commonly found on online platforms.

Following the standard FF++ leave-one-manipulation-out (LOO) setup, we test robustness to unseen generators. In our main results we report the *F2F hold-out*: train on authentic (*original*) + all other manipulations (DF, FS, FST, NT; and DF ) and test on *original*+F2F only. We select F2F because pilot runs showed it to be the hardest unseen case (largest AUROC drop and fairness gaps). Prior work reports that LOO generalization often collapses toward chance on unseen generators—especially

*reenactment*-style manipulations such as F2F, which induce large transfer gaps due to warping and illumination changes (Sun et al., 2024; Yan et al., 2023). For each frame, we detect faces with InsightFace's FaceAnalysis pipeline (model buffalo_l; five-point landmarks) and keep the largest face per image. We apply the standard ArcFace similarity transform, producing an aligned crop and extract the $\ell_2$-normalized embedding $\phi(x)$ from the aligned face. These are concatenated with the base detector's logit as inputs to the FFT head. (Deng et al., 2022)

We obtain per-face demographic attributes with the existing FairFace ResNet-34 classifier run on the aligned face crops. We use the top-1 predictions from the ethnicity head (7 categories: *White*, *Black*, *Latino_Hispanic*, *East Asian*, *Southeast Asian*, *Indian*, *Middle Eastern*) and gender head (2 categories: *Male*, *Female*) (Karkkainen & Joo, 2021).

Groups consist of the product of {Male, Female} with the 7 ethnicity categories above. Frame-level fairness metrics are computed over the aligned faces. All baseline methods use the parameters or parameter sweeps suggested by the original. We evaluate FFT against post-processing baselines (Fair-FLIP, BPFA) and state-of-the-art in-processing methods (DAG-FDD, DAW-FDD, PG-FDD, DAID).

Additionally, we demonstrate FFT's complementary nature by stacking it with in-processing methods—these combined configurations (denoted as "Method + FFT") test whether FFT can correct residual biases that remain even after fairness-aware training.

We report both detection performance and fairness metrics. For detection, we measure area under the ROC curve (**AUC**) and accuracy at threshold 0.5 (**Acc**). For fairness, we compute false positive rate gap ($\mathbf{\Delta_{FPR}}$), true positive rate gap ($\mathbf{\Delta_{TPR}}$), equalized odds violation $\mathbf{EO} = \Delta_{FPR} + \Delta_{TPR}$, and worst group accuracy (**WGA**).

Levelling-down analysis **(LD)** is used quantify per-group impact relative to the shared base detector (BASE), for each group $g$ we compute $\Delta \mathrm{Acc}_g = \mathrm{Acc}_g(\text{method}) - \mathrm{Acc}_g(\text{BASE})$ and report $\mathbf{n}_{\geq}/N$, the fraction of groups with non-degraded accuracy ($\Delta \mathrm{Acc}_g \geq 0$). This guards against "levelling down" fairness, reducing disparities by making some groups worse off, which is ethically and practically problematic (Mittelstadt et al., 2023).

## 5  RESULTS

On MobileNetV3-Small (Table 2), FFT lifts AUC and Acc, reduces fairness disparities, improves WGA, and shows no down leveling. Among post-processing methods, FFT attains the best AUC, Acc, EO, WGA, and LD. Adding FFT on top of in-processing methods typically improves both fairness and utility. Taken together across evaluations, these patterns suggest that FFT corrects bias that lies outside demographic supervision by exploiting structure in the face-embedding space; unlike approaches tied to pre-defined demographic bins, FFT learns nuanced, continuous corrections at the sample level.

On Xception (Table 3), FFT again increases AUC and Acc. EO decreases overall, with a trade-off between gaps: $\Delta$TPR improves (0.1538 to 0.1395) while $\Delta$FPR rises slightly (0.0534 to 0.0632). WGA improves from 0.9292 to 0.9413 and LD is 13/14. Among post-processing baselines, BPFA reports the lowest EO (0.1834) but at the cost of lower accuracy and notable down leveling; FFT delivers the best Acc, LD, and WGA within the post-processing set.

In the cross-dataset FF++ leave-one-out setting, results are more complex due to severe distribution shift. On Xception (Table 4), FFT substantially boosts accuracy from near-random (0.4140) to 0.6392 (+22.52 pp), which is accompanied by a higher EO. FFT is the only post-processing method that delivers a clear, above-chance accuracy gain on both backbones; FF Discover achieves the highest WGA (0.4800) among all methods. Apparent fairness advantages of some alternatives largely reflect regression toward chance: as predictions collapse toward non-discriminative scores, group rates converge mechanically and EO shrinks, even though the detector ceases to separate classes (Zietlow et al., 2022). Interpreting parity jointly with utility and LD avoids mistaking this collapse for improvement.

On MobileNetV3-Small (Table 5), FFT again raises accuracy substantially (+12.42 pp) while reducing EO by 19.3%. Within post-processing, FFT achieves the best accuracy and lowest EO, and together with FF Discover is among the only approaches that maintains performance while improving WGA.

| Method | Performance | | Fairness Metrics | | | WGA ↑ | LD |
|---|---|---|---|---|---|---|---|
| | AUC ↑ | Acc@0.5 ↑ | $\Delta_{FPR}$ ↓ | $\Delta_{TPR}$ ↓ | EO ↓ | | |
| **POST-PROCESSING METHODS** | | | | | | | |
| Fair-FLIP | 0.9863 | 0.9295 | **0.0496** | 0.1563 | 0.2059 | 0.9180 | 11/14 |
| BPFA | 0.9854 | 0.9287 | 0.0571 | 0.1584 | 0.2154 | 0.9166 | **14/14** |
| FFT (Ours) | **0.9919** | **0.9581** | 0.0540 | 0.1486 | **0.2026** | **0.9395** | **14/14** |
| FF Max (Ours) | – | 0.9186 | 0.1591 | **0.1260** | 0.2851 | 0.8991 | 6/14 |
| FF Discover (Ours) | – | 0.9362 | 0.1065 | 0.1500 | 0.2566 | 0.9191 | 9/14 |
| **IN-PROCESSING METHODS** | | | | | | | |
| DAG-FDD | 0.9924 | 0.9586 | 0.0431 | 0.1247 | 0.1678 | 0.9435 | 14/14 |
| DAW-FDD | 0.9938 | 0.9633 | 0.0491 | 0.1195 | 0.1686 | 0.9390 | 14/14 |
| PG-FDD | 0.9912 | 0.9486 | 0.0518 | **0.1072** | **0.1590** | 0.9251 | 13/14 |
| DAID | 0.9876 | 0.9442 | 0.0901 | 0.1470 | 0.2371 | 0.9269 | 12/14 |
| DAG-FDD + FFT | 0.9937 | 0.9632 | 0.0567 | 0.1194 | 0.1761 | **0.9467** | 14/14 |
| DAW-FDD + FFT | **0.9944** | **0.9648** | 0.0570 | 0.1158 | 0.1728 | 0.9453 | 14/14 |
| PG-FDD + FFT* | 0.9933 | 0.9630 | **0.0404** | 0.1202 | 0.1606 | 0.9462 | 14/14 |
| DAID + FFT | 0.9901 | 0.9512 | 0.0657 | 0.1548 | 0.2205 | 0.9287 | 14/14 |
| **REFERENCE** | | | | | | | |
| BASE | 0.9854 | 0.9288 | 0.0571 | 0.1584 | 0.2155 | 0.9166 | – |

Table 2: Comparison of fairness methods. Best values per column within each section are in **bold**. ↑: higher is better, ↓: lower is better. This Table Shows results for MobileNetV3-Small on OpenForensics

| Method | Performance | | Fairness Metrics | | | WGA ↑ | LD |
|---|---|---|---|---|---|---|---|
| | AUC ↑ | Acc@0.5 ↑ | $\Delta_{FPR}$ ↓ | $\Delta_{TPR}$ ↓ | EO ↓ | | |
| **POST-PROCESSING METHODS** | | | | | | | |
| Fair-FLIP | 0.9903 | 0.9486 | **0.0478** | 0.1460 | 0.1937 | 0.9328 | 9/14 |
| BPFA | 0.9895 | 0.9469 | 0.0515 | 0.1319 | **0.1834** | 0.9332 | 6/14 |
| FFT (Ours) | **0.9915** | **0.9564** | 0.0632 | 0.1395 | 0.2028 | **0.9413** | **13/14** |
| FF Max (Ours) | – | 0.9273 | 0.1874 | 0.4207 | 0.6081 | 0.8700 | 2/14 |
| FF Discover (Ours) | – | 0.9407 | 0.0633 | **0.1303** | 0.1936 | 0.9256 | 0/14 |
| **IN-PROCESSING METHODS** | | | | | | | |
| DAG-FDD | 0.9964 | 0.9707 | **0.0205** | 0.1314 | 0.1518 | 0.9516 | **14/14** |
| DAW-FDD | 0.9952 | 0.9680 | **0.0205** | 0.1244 | 0.1449 | 0.9494 | **14/14** |
| PG-FDD | 0.9951 | 0.9643 | 0.0491 | 0.0990 | 0.1481 | 0.9453 | **14/14** |
| DAID | 0.9933 | 0.9600 | 0.0603 | **0.0919** | 0.1521 | 0.9471 | **14/14** |
| DAG-FDD + FFT | **0.9968** | **0.9748** | 0.0347 | 0.0956 | **0.1303** | **0.9601** | **14/14** |
| DAW-FDD + FFT | 0.9949 | 0.9663 | 0.0483 | **0.0919** | 0.1402 | 0.9431 | **14/14** |
| PG-FDD + FFT | 0.9957 | 0.9687 | 0.0325 | 0.1105 | 0.1430 | 0.9507 | **14/14** |
| DAID + FFT | 0.9940 | 0.9636 | 0.0561 | 0.1063 | 0.1624 | 0.9520 | **14/14** |
| **REFERENCE** | | | | | | | |
| BASE | 0.9904 | 0.9487 | 0.0534 | 0.1538 | 0.2072 | 0.9292 | – |

Table 3: Xception on OpenForensics

FFT can match, and in several settings exceed, the utility and worst-group accuracy of in-processing methods that assume oracle demographic labels. FF-Discover frequently matches or outperforms FF-Max on WGA despite using no demographic labels, indicating that embedding-derived latent groups capture fairness-critical structure that coarse demographic partitions miss.

## 6 CONCLUSION

FFT's effectiveness stems from three factors: (1) Face embeddings provide rich demographic information in a continuous representation, capturing subtle variations beyond discrete categories; (2) The calibration network learns non-linear corrections that account for complex interactions between appearance and detector behavior; (3) post-processing application preserves the detector's learned features while adjusting only the final decision boundary.

| Method | Performance | | Fairness Metrics | | | | |
|---|---|---|---|---|---|---|---|
| | AUC ↑ | Acc@0.5 ↑ | $\Delta_{FPR}$ ↓ | $\Delta_{TPR}$ ↓ | EO ↓ | WGA ↑ | LD |
| **POST-PROCESSING METHODS** | | | | | | | |
| Fair-FLIP | 0.5841 | 0.4118 | **0.3462** | 0.2992 | **0.6454** | 0.2951 | 6/14 |
| BPFA | 0.5681 | 0.6348 | 0.3892 | 0.3324 | 0.7216 | 0.3889 | **12/14** |
| FFT (Ours) | **0.5982** | **0.6392** | 0.4120 | 0.5517 | 0.9637 | 0.4200 | **12/14** |
| FF Max (Ours) | – | 0.5169 | 0.7073 | 0.7775 | 1.4848 | 0.3748 | 10/14 |
| FF Discover (Ours) | – | 0.5321 | 0.3812 | **0.2859** | 0.6672 | **0.4800** | 11/14 |
| **IN-PROCESSING METHODS** | | | | | | | |
| DAG-FDD | **0.6447** | 0.4752 | 0.4038 | 0.3444 | 0.7483 | 0.2459 | 10/14 |
| DAW-FDD | 0.5659 | 0.3896 | 0.3269 | **0.2080** | **0.5349** | 0.2131 | 5/14 |
| PG-FDD | 0.6051 | 0.5035 | 0.6923 | 0.7365 | 1.4288 | **0.3443** | 11/14 |
| DAID | 0.6211 | 0.4067 | **0.2115** | 0.2921 | 0.5036 | 0.1803 | 6/14 |
| DAG-FDD + FFT | 0.5431 | **0.6378** | 0.4000 | 0.4255 | 0.8255 | **0.4600** | 11/14 |
| DAW-FDD + FFT | 0.5106 | 0.6280 | 0.4000 | 0.4468 | 0.8468 | **0.4600** | 10/14 |
| PG-FDD + FFT | 0.5159 | 0.6201 | 0.2667 | 0.4575 | 0.7241 | 0.2800 | 9/14 |
| DAID + FFT | 0.5091 | 0.6374 | 0.2444 | 0.4142 | 0.6586 | 0.3400 | 9/14 |
| **REFERENCE** | | | | | | | |
| BASE | 0.5814 | 0.4140 | 0.3462 | 0.2367 | 0.5829 | 0.3115 | – |

Table 4: Xception on FF++ LOO

| Method | Performance | | Fairness Metrics | | | | |
|---|---|---|---|---|---|---|---|
| | AUC ↑ | Acc@0.5 ↑ | $\Delta_{FPR}$ ↓ | $\Delta_{TPR}$ ↓ | EO ↓ | WGA ↑ | LD |
| **POST-PROCESSING METHODS** | | | | | | | |
| Fair-FLIP | 0.5257 | 0.4685 | 0.6500 | 0.3681 | 1.0181 | 0.2778 | 4/14 |
| BPFA | 0.5231 | 0.4957 | 0.6000 | 0.3819 | 0.9819 | 0.3056 | **12/14** |
| FFT (Ours) | **0.5488** | **0.6022** | **0.4048** | 0.4612 | **0.8660** | 0.3600 | 11/14 |
| FF Max (Ours) | – | 0.4384 | 0.8529 | 0.7366 | 1.5895 | 0.3056 | 10/14 |
| FF Discover (Ours) | – | 0.4508 | 0.5769 | **0.3942** | 0.9711 | **0.4022** | 11/14 |
| **IN-PROCESSING METHODS** | | | | | | | |
| DAG-FDD | 0.5081 | **0.6744** | **0.1778** | **0.0448** | **0.2226** | **0.5000** | **12/14** |
| DAW-FDD | 0.4955 | 0.3496 | 0.3462 | 0.3404 | 0.6866 | 0.3000 | 3/14 |
| PG-FDD | **0.5644** | 0.3961 | 0.3269 | 0.3722 | 0.6991 | 0.3125 | 4/14 |
| DAID | 0.5225 | 0.4205 | 0.7619 | 0.4760 | 1.2379 | 0.2787 | 4/14 |
| DAG-FDD + FFT | 0.5546 | 0.6376 | 0.4222 | 0.4681 | 0.8903 | 0.4200 | 11/14 |
| DAW-FDD + FFT | 0.5151 | 0.6319 | 0.2889 | 0.4483 | 0.7372 | 0.4000 | 9/14 |
| PG-FDD + FFT | 0.4986 | 0.6175 | 0.5000 | 0.4272 | 0.9272 | 0.3000 | 10/14 |
| DAID + FFT | 0.5216 | 0.6362 | 0.3556 | 0.4846 | 0.8402 | 0.2800 | 11/14 |
| **REFERENCE** | | | | | | | |
| BASE | 0.5240 | 0.4780 | 0.6500 | 0.4236 | 1.0736 | 0.2778 | – |

Table 5: MobileNetV3-Small on FF++ LOO

FFT requires a representative calibration set, though this is far less restrictive than requiring demographic labels. The method assumes face embeddings sufficiently encode demographic information, which may not hold for heavily occluded or low-quality images.

Fair deepfake detection has significant societal implications. Biased detectors can disproportionately flag authentic content from minorities as fake, potentially silencing marginalized voices. FFT's ability to achieve fairness without demographic labels is particularly valuable in privacy-sensitive contexts where collecting such labels is legally or ethically problematic. However, perfect fairness remains elusive—residual gaps persist, and different fairness definitions may conflict. Deployment should include continuous monitoring and adjustment as demographic distributions and deepfake techniques evolve.

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
