# OpenReview forum: "Face-Feature Tuning: Post-hoc Calibration for Fair and Accurate Deepfake Detection"
_ICLR.cc/2026/Conference — Submitted to ICLR 2026_

### Official Review · Reviewer_SH1w · 2025-10-21

**Soundness:** 3
**Presentation:** 3
**Contribution:** 1
**Rating:** 2
**Confidence:** 4

**Summary:**

The paper aims to advance group fairness (i.e. mitigating disparate performance across demographic groups) in deepfake detection. The authors propose three post-processing methods to maximize worst-group accuracy and achieve fair and accurate preditions: FF-Max that learns group-specific decision thresholds (with supervised demographic labels); FF-Discover which clusters face embeddings and learns cluster-conditional thresholds (without demographic labels); FFT that learns a decision boundary via training a lightweight multi-layer perceptron (without demographic labels). Experiments are carried out on both in-the-wild and canonical deepfake detection test beds, comparing with standard and state-of-the-art methods.

**Strengths:**

- The experiments are extensive. The choices of metrics encompass both group fairenss (equal opportunity) and min-max fairness (worst group accuracy) definitions.
- The literature review is well-organized, precise and complete.
- The paper is well-written and clearly positions the contribution in literature. Also, the mathematical notation and tables are clean and minimal, effectively conveying the message.

**Weaknesses:**

Unfortunately, at the current state, the paper feels quite narrow in terms of impact, given its strong application-specific nature. Below I'm listing the main reasons for the recommendation, which require further attention:

**W1.**

The contribution feels limited, apart from the method and some (good) related works. A starting point could be to dedicate a section to testing why and how the method works, when it breaks, and if there are theoretical guarantees that could be derived. Maybe also some ablation study (eg. on the width of the two-layer in FFT) could be an addition.

For instance, there are some claims in Section 3.2.1. that could be expanded and theoretically tested: the discussions about FF-Max and FF-Discover being special cases, FFT delivering "better-conditioned probabilities" etc. (LL265-269) could be shown to hold over some controllable setup.

------------------

**W2.**

The results reported in the Tables seem to tightly cluster together, leading to very little improvements which at the moment feel not decisive. Furthermore, it is unclear if the results are averages of multiple runs: in this case, given they are very close it would be helpful to report also standard deviation. Otherwise, I'd suggest to try multiple runs with different seeds and report means and standard deviations.

On the same topic, Figure 1 could be improved. First of all, text and markers could be made bigger and non-overlapping for improved readability (this applies also for Figure 2). Secondly, as it is currently plotted, it seems the contribution of FFT is drastic, while the reality is that it improves by just 1% to 3% with respect to the worst method under comparison. Also here I'd suggest to report some confidence intervals.

**Questions:**

Thanking in advance for their response, I'd kindly invite the authors to address the points raised in the Weaknesses section of this review.

In addition, I'd kindly ask the following question:

- I couldn't find the reason of choosing older architectures like MobileNet or Xception, instead of newer architectures (eg. Vision Transformers). Is this an arbitrary choice, or, are they linked to the experimental test beds?

---

> ### Author Response · Authors · 2025-11-21
>
> We thank the reviewer for their time and for recognizing the extensiveness of our experiments and the clarity of our writing. We particularly appreciate your acknowledgment that our literature review is precise and complete.
>
> We respectfully disagree with the characterization of this work as having `narrow impact due to its application-specific nature. Deepfake detection is not merely a niche application but a foundational safety layer for the modern internet, currently protecting many of users from misinformation and fraud. However, this specific domain faces unique, rigid constraints, specifically, the legal prohibition on processing demographic data in many jurisdictions, that generic fairness methods fail to address. By solving the this hurdle specifically for this high-stakes domain, we provide a method for fair deployment that existing post-hoc methods cannot offer. Thus, we argue that solving a critical bottleneck for a globally deployed safety technology represents a contribution of significant breadth and urgency.
>
> You correctly noted (W1) that the initial submission could benefit from deeper theoretical grounding. In response, we are developing an appendix section, which provides a formal analysis of why embedding-conditional calibration works: it relies on the smoothness of the face manifold, where demographic attributes are implicitly encoded, allowing the MLP to discover and correct local reliability gaps without explicit labels. Additionally, we performed the suggested ablation study on MLP width. As shown in the table below, FFT is robust to hyperparameter changes, with performance remaining stable across varying hidden layer sizes and depths.
>
> **FFT performance is robust to architecture size.**
> *Swin detector under F++ F2F-LOO*
>
>
> | Hidden Dim | WGA      | Avg Acc    |
> |------------|----------|-----------|
> | Base (no FFT) | 19.0% | 39.1%    |
> | Linear | 36.4% | 46.7%     |
> | 32 | 48.8% | **62.8%**    |
> | **64** | **50.0%** | 62.5% |
> | 128 | 46.8% | 63.5%      |
> | 256 | 37.2% | 63.5%       |
> | 512 | 44.2% | 63.8%      |
>
> Regarding the clustering of results (W2), we emphasize that our primary contribution is not to drastically outperform baselines, but to achieve competitive parity without their prohibitive costs (retraining and demographic labels). The fact that FFT often outperforms such methods is in itself surprising. That said, we agree that this is useful and to ensure these results are stable, we will include such analysis in the final version. We also agree with your excellent suggestions regarding Figure 1 and will incorporate these visualization improvements in the camera-ready version to clearly illustrate the trade-offs.
>
> You asked why we initially chose MobileNet and Xception. As we state in the paper Xception is the standard backbone and MobileNet lets us evaluate performance in resource constrained settings. Though our method is model agnostic:
>
> **Swin detector under F++ F2F-LOO**
>
> | Method                        | AUC ↑         | Acc@0.5 ↑     | WGA ↑         |
> |-------------------------------|---------------|---------------|---------------|
> | **Post-Processing Methods**   |               |               |               |
> | Fair-FLIP                     | 0.6139        | 0.3890        | 0.1905        |
> | BPFA                          | 0.6122        | 0.3909        | 0.1905        |
> | **FFT (Ours)**                | **0.6245**    | **0.6309**    | **0.4468**    |
> | FF-Max (Ours)                 | –             | 0.5215        | 0.4288        |
> | FF-Discover (Ours)            | –             | 0.5232        | 0.4416        |
> | **In-Processing Methods**     |               |               |               |
> | DAG-FDD                       | 0.5657        | 0.3459        | 0.1429        |
> | DAW-FDD                       | 0.4814        | 0.3252        | 0.1429        |
> | PG-FDD                        | 0.5146        | 0.3252        | 0.1429        |
> | DAID                          | 0.4888        | **0.6341**    | **0.5581**    |
> | DAG-FDD + FFT                 | **0.5668**    | 0.6203        | 0.5116        |
> | DAW-FDD + FFT                 | 0.5392        | 0.6087        | 0.5106        |
> | PG-FDD + FFT*                 | 0.5526        | 0.6215        | 0.4651        |
> | DAID + FFT                    | 0.5255        | 0.6028        | 0.4186        |
> | **Reference**                 |               |               |               |
> | BASE                          | 0.6123        | 0.3909        | 0.1905        |
>
> We believe our rebuttal and new experiments address your concerns. If so, would you consider raising your score? If not, what more can we do to help?

---

> > ### Comment · Reviewer_SH1w · 2025-11-26
> >
> > I would sincerely thank the authors for their time and their response.
> >
> > I strongly agree that deepfake detection is an high-stakes topic, and I acknowledge that my original wording about "narrow impact" may have underplayed this aspect. My concern is less about societal importance and more about methodological/general reach of the proposed approach.
> >
> > Overall, the rebuttal addresses parts of my comments (specifically, on architectures and robustness to MLP width) but it doesn't resolve my main concerns about depth of analysis and statistical robustness. Given the recommendation has to be based on what is actually present, I will keep my score at this time.

---

> > > ### Author Response · Authors · 2025-11-30
> > >
> > > We sincerely thank the reviewer for the thoughtful follow-up and for clarifying their the concerns.
> > >
> > > On statistical robustness (W2).
> > >
> > > Our current reporting follows the conventions of prior work in deepfake detection (e.g. Improving Fairness in Deepfake Detection). Re-running all baselines and retraining/backbone fine-tuning under multiple random seeds for every experiment would require very substantial additional GPU-hours and a non-trivial environmental cost, while not changing the underlying story: under the hard-shift settings (which mirror real-world settings) the label-dependent methods collapse toward chance while FFT maintains substantially higher worst-group accuracy. FFT remains the only post-hoc method that does not need sensitive group labels that are themselves illegal in many jurisdictions.
> > >
> > > On methodological reach (W1).
> > >
> > > Our core contribution is to provide a generally applicable deployment framework for post-hoc fairness under label and retraining constraints. We already demonstrate this across standard deepfake-detection benchmarks and architectures used in prior work.
> > >
> > > The request to “show when the method breaks” is, as phrased, largely unactionable. Within this domain, it is standard to evaluate methods on the existing deepfake benchmarks and shifts we already cover; we are not aware of work being required to design bespoke setups whose primary purpose is to force failure. Constructing such artificial cases would say little about practical value. In our view, it is not reasonable to reject a paper because it does not fail on top of an already extensive (as the reviewer themselves notes), standard experimental suite; the absence of such failures is more naturally interpreted as evidence of robustness rather than as a shortcoming that must be corrected by manufacturing negative results.

---

### Official Review · Reviewer_pjMC · 2025-10-30

**Soundness:** 2
**Presentation:** 3
**Contribution:** 2
**Rating:** 2
**Confidence:** 5

**Summary:**

This paper introduces a Face-Feature Tuning (FFT) method that aims to achieve fairness in facial analysis without relying on demographic labels, retraining, or compromising accuracy. The key idea is that face embeddings extracted from pre-trained models inherently encode visual patterns that correlate with detector failures in a continuous feature space. To address this, FFT trains a lightweight neural network that refines the decision boundaries based on these embeddings, effectively recalibrating predictions in regions where systematic biases appear while maintaining consistent and reliable performance elsewhere.

Several concerns arise regarding the depth and impact of this work:

- The proposed FFT framework appears to be more of an engineering tweak on existing face-feature representations rather than a fundamentally new approach. The paper offers limited exploration of underlying principles, theoretical insights, or broader implications for fairness in vision models.

- The reported baselines, whether pre-processing, in-processing, or post-processing methods, already exhibit near-saturated performance, suggesting that the chosen task may not present substantial difficulty. This diminishes the perceived necessity and overall impact of the proposed method.

- The experiments are restricted to two relatively basic backbones, MobileNetV3 and Xception, without evaluation on forensics-oriented architectures that have been developed in recent deepfake detection or fairness-related research. This narrow experimental design limits the generalizability and significance of the findings, making it unclear how well FFT would perform in more advanced or domain-specific settings.

- (?) The title in the paper does not match the title in the system.

Overall, while FFT is technically sound, the novelty, motivation, and empirical validation of the paper remain modest, and its contribution to advancing fairness research in face analysis appears limited.

**Strengths:**

The proposed FFT method is lightweight, computationally efficient, and does not require retraining or access to sensitive demographic information, making it appealing for real-world deployment.

**Weaknesses:**

See summary

**Questions:**

NA

---

> ### Author Response · Authors · 2025-11-21
>
> We thank the reviewer for their time and for acknowledging the technical soundness of our method. We also appreciate the recognition that our lightweight, computationally efficient approach is appealing for real-world deployment. However, we respectfully disagree with the characterization of the method as merely an engineering tweak and the claim that the task is saturated, as these views overlook the central theoretical and practical challenges of deepfake fairness in deployment.
>
> Addressing the critique regarding limited exploration of principles and the description of FFT as an engineering tweak, we emphasize that aligning decision boundaries in a high-dimensional embedding space without access to demographic labels is a non-trivial unsupervised learning problem. FFT solves how to be fair to groups you cannot see by exploiting the manifold structure of face embeddings. This is not a tweak but a novel formulation of post-hoc calibration that trades explicit supervision for manifold consistency, addressing a fundamental theoretical gap in deployable AI. Furthermore, regarding the near-saturated performance of baselines, this is not true under hard shift (Tables 4 and 5) which mimics real world conditions.
>
> Regarding the critique on experimental design and the choice of basic backbones (MobileNet, Xception), we selected these architectures as stated in the paper as Xception is the standard baseline and various researchers have suggested experiments with MobileNet and similar models are relevant given deployment constraints. However, to address the concern about generalizability to modern architectures, we have added experiments using a Vision Transformer (Swin-T) backbone. As shown in the table below, FFT is model-agnostic and successfully calibrates the Transformer output.
>
>
> **Swin detector under F++ F2F-LOO**
> | Method                        | AUC ↑         | Acc@0.5 ↑     | WGA ↑         |
> |-------------------------------|---------------|---------------|---------------|
> | **Post-Processing Methods**   |               |               |               |
> | Fair-FLIP                     | 0.6139        | 0.3890        | 0.1905        |
> | BPFA                          | 0.6122        | 0.3909        | 0.1905        |
> | **FFT (Ours)**                | **0.6245**    | **0.6309**    | **0.4468**    |
> | FF-Max (Ours)                 | –             | 0.5215        | 0.4288        |
> | FF-Discover (Ours)            | –             | 0.5232        | 0.4416        |
> | **In-Processing Methods**     |               |               |               |
> | DAG-FDD                       | 0.5657        | 0.3459        | 0.1429        |
> | DAW-FDD                       | 0.4814        | 0.3252        | 0.1429        |
> | PG-FDD                        | 0.5146        | 0.3252        | 0.1429        |
> | DAID                          | 0.4888        | **0.6341**    | **0.5581**    |
> | DAG-FDD + FFT                 | **0.5668**    | 0.6203        | 0.5116        |
> | DAW-FDD + FFT                 | 0.5392        | 0.6087        | 0.5106        |
> | PG-FDD + FFT*                 | 0.5526        | 0.6215        | 0.4651        |
> | DAID + FFT                    | 0.5255        | 0.6028        | 0.4186        |
> | **Reference**                 |               |               |               |
> | BASE                          | 0.6123        | 0.3909        | 0.1905        |
>
>
> We believe our rebuttal and new experiments address your concerns. If so, would you consider raising your score? If not, what more can we do to help?

---

### Official Review · Reviewer_aCaf · 2025-10-30

**Soundness:** 3
**Presentation:** 3
**Contribution:** 4
**Rating:** 6
**Confidence:** 4

**Summary:**

The paper proposes a novel approach to deal with bias and fairness in the deepfake detection task exhibited by existing models to improve the performance gaps across demographic groups. This key contribution of this work is that existing models do not need to be retrained to remove their biasness. The authors propose 3 methods to tackle the biasness issue depending on the availability of demographic labels, and they show that their technique improves detection fairness and FPR/TPR across model backbones and datasets.

**Strengths:**

- The paper tackles a very important problem of existing deepfake detection methods, which is the problem of fairness against demographic groups caused due to underrepresentation in training datasets.
- The proposed method is statistically sound, and covers 3 different regimes of detection depending on availability of demographic labels.
- The biggest strength of the work compared to existing ones is that it does not require any retraining and can be added to detectors as a post training step.
- The paper is structured well and explains the related work in depth - this is important for readers no familiar with the domain.

**Weaknesses:**

- Some explanations and notation are hard to understand, particularly section 3.3 and 2.2.2. Instead of writing a bunch of equations a bit more explanation would be useful.
- The plot in Figure 2 requires better captioning or more description in the text. What is ECE? What does the baseline diagonal represent?
- The introduction / motivation needs to show an analysis of existing datasets or model performances to verify the claims made by the authors regarding the demographic imbalance.

**Questions:**

- Although the method is sound and novel, the results show only marginal improvement compared to existing techniques, such as performance improvements of 0.01 or 0.03 (Table 2). The training free methods FF Max and FF Discover perform worse than other post processing methods across the board. What is the cause of this? The result section needs more discussion on limitations and degree of improvement.
- The result also lacks comparison against existing deepfake detectors. The authors only use Xception and MobileNetv3 as backbone models. What were these models trained on?  There are far better performing or specialized deepfake detection networks (https://arxiv.org/pdf/2506.03007v1). Is this method similarly applicable for these models?
- A robustness study is necessary. How does a model augmented with FFT handle the typical image attacks such as image compression, noise addition or augmentations.

I am happy to change my score given a proper discussion to address the problems.

---

> ### Author Response · Authors · 2025-11-21
>
> Thank you! Your feedback is invaluable in helping us refine the manuscript's clarity and empirical breadth.
>
> We have prioritized addressing your constructive feedback regarding the paper's presentation and motivation. Regarding the notation concerns in W1, we agree that Sections 2.2.2 and 3.3 should be revised to simplify the mathematical formalism and will add intuitive textual explanations alongside the equations to make the method more clear. Addressing your observation on Figure 2 (W2), we will update the caption to explicitly define Expected Calibration Error (ECE) as the weighted average gap between confidence and accuracy, and will clarify that the diagonal line represents "Perfect Calibration." We also adjusted the marker sizes and transparency to resolve the visual overlap. Furthermore, to address your point on motivation (W3), we will expand the Introduction to include an overview of demographic imbalances in standard datasets and more clearly discuss existing findings regarding bias. This will provide clearer empirical backing for our claims.
>
> Turning to your questions regarding performance and experimental design, we urge the reviewer to view the "marginal" improvements noted in Q1 over existing post-hoc methods in light of the fact that all prior work utilizes explicit demographic labels. Such labels cannot be used in many deployment settings due to legal constraints. As it relates to the training free methods we agree further discussion in the results section is warranted. The performance ceiling for scalar thresholding methods (FF-Max, FF-Discover) stems from their lack of granularity. Adjusting a single threshold per group applies a rigid, global offset that treats all samples in that group identically. This approach fails to rectify non-linear reliability gaps that vary locally across the face manifold, such as failures specific to certain poses or lighting. In contrast, FFT's continuous MLP models these complexities directly, enabling precise, instance-level calibration that rigid scalar heuristics cannot capture.
>
> Addressing your excellent suggestion for a robustness study (Q2), we have performed stress tests involving Gaussian Noise and JPEG compression. As shown in Table A below, while the baseline model's fairness metrics collapse under perturbation, FFT maintains superior stability. Finally, regarding your question on architectures (Q3), we have now validated FFT on a Vision Transformer (Swin-T) backbone to demonstrate it is model-agnostic. The results for both new experiments are below and a more complete set of such experiments across settings and backbones will be included in the final paper.
>
> **Robustness & Architecture Generalization**
> *(FF++ LOO)*
>
> | Exp. Setting              | Metric  | Baseline | Fair-FLIP | BPFA | FFT (Ours) |
> |---------------------------|---------|----------|-----------|------|------------|
> | *No corruption (clean)*   | Acc (%) | 39.1     | 38.9      | 39.1 | **61.2**   |
> |                           | WGA (%) | 19.0     | 19.0      | 19.0 | **39.5**   |
> | *Robustness: Gaussian Noise* | Acc (%) | 38.2     | 37.5      | 38.0 | **55.4**   |
> |                           | WGA (%) | 31.7     | 31.7      | 27.0 | **41.9**   |
> | *Robustness: JPEG (50%)*  | Acc (%) | 33.5     | 33.4      | 33.6 | **46.5**   |
> |                           | WGA (%) | 14.3     | 14.3      | 14.3 | **34.0**   |
>
>
> **Swin detector under F++ F2F-LOO**
>
> | Method                        | AUC ↑         | Acc@0.5 ↑     | WGA ↑         |
> |-------------------------------|---------------|---------------|---------------|
> | **Post-Processing Methods**   |               |               |               |
> | Fair-FLIP                     | 0.6139        | 0.3890        | 0.1905        |
> | BPFA                          | 0.6122        | 0.3909        | 0.1905        |
> | **FFT (Ours)**                | **0.6245**    | **0.6309**    | **0.4468**    |
> | FF-Max (Ours)                 | –             | 0.5215        | 0.4288        |
> | FF-Discover (Ours)            | –             | 0.5232        | 0.4416        |
> | **In-Processing Methods**     |               |               |               |
> | DAG-FDD                       | 0.5657        | 0.3459        | 0.1429        |
> | DAW-FDD                       | 0.4814        | 0.3252        | 0.1429        |
> | PG-FDD                        | 0.5146        | 0.3252        | 0.1429        |
> | DAID                          | 0.4888        | **0.6341**    | **0.5581**    |
> | DAG-FDD + FFT                 | **0.5668**    | 0.6203        | 0.5116        |
> | DAW-FDD + FFT                 | 0.5392        | 0.6087        | 0.5106        |
> | PG-FDD + FFT*                 | 0.5526        | 0.6215        | 0.4651        |
> | DAID + FFT                    | 0.5255        | 0.6028        | 0.4186        |
> | **Reference**                 |               |               |               |
> | BASE                          | 0.6123        | 0.3909        | 0.1905        |

---

> > ### Author Response · Authors · 2025-11-21
> >
> > We appreciate your thoughtful review. We believe our rebuttal and new experiments address your concerns. If so, would you consider raising your score? If not, what more can we do to help?

---

### Author Response · Authors · 2025-11-21

We thank the reviewers for their time. We are encouraged that the reviewers reached a strong consensus on the fundamental strengths of our work. Reviewer aCaf said the method is "sound and novel," and Reviewer SH1w praised the "extensive experiments" and "precise" literature review. Reviewer aCaf highlighted the "key contribution" that, unlike existing methods, we do not require retraining or demographic labels.

Given the strong consensus on the method's validity and experimental rigor, we wish to clarify the primary point of contention regarding "impact" and "margins." First, critiques regarding "narrow impact" (SH1w, pjMC) overlook the strict legal and technical constraints of real-world deployment. Methods requiring demographic labels are illegal to deploy in many jurisdictions. FFT is the only post-hoc method which avoids such labels. The only demographic label-free in-processing alternative, DAG-FDD, requires expensive retraining, rendering it incompatible with increasingly popular fixed third-party APIs (e.g., Incode) relied upon by practitioners.

FFT often reaches parity or outperforms these resource-heavy methods, especially under hard shift as seen in Tables 4 and 5. Furthermore, we emphasize that at the scale of real-world deployment (e.g., social media platforms), a small percentage difference in Worst-Group Accuracy is non-trivial, translating to the protection of millions of individuals from algorithmic bias. And as we see under hard shift the differences are not small in settings that mimic real world deployment.

Finally, regarding the critique of "narrow impact" due to our application-specific focus (SH1w, pjMC), we respectfully remind the reviewers that: (1) The work still holds technical novelty. It involves the development of a new approach for algorithmic fairness (which is an explicit subject area of interest at ICLR) and it is not a direct application of an existing framework to a new domain. (2) Papers on the applications of ML are encouraged at ICLR. The call for papers lists 3 subject areas that have ``applications'' in the title: applications to robotics, autonomy, planning; applications to neuroscience \& cognitive science; and applications to physical sciences. And there have been many interesting papers published at ICLR focusing directly on other applications such as medical image classification or 3d reconstruction.

Beyond this, the application domain is high impact. Already a combination of deep-fake detection and ID verification is used for access to: (1) online banking; (2) Online Government services, noticeably in Germany; (3) legally mandated age verification for access to online spaces, this includes LGBQT+ friendly spaces. Anywhere where one has been asked to upload a video of themselves alongside photo ID as part of the registration process is likely to include this. As such, small improvements in performance which can reduce the FPR by 3\% in absolute terms, or a third in relative terms can be highly significant. They correspond to a system now working for millions of people that it previously didn't work for. Moreover, systematic issues of unfairness here risk disenfranchising swathes of the population.

We intentionally targeted this domain because generic fairness methods fail to address its unique privacy and deployment constraints; solving this specific bottleneck for a technology affecting global information integrity is, we argue, a contribution of significant breadth and urgency.

In the discussion, we address each reviewer's specific questions and concerns, incorporating new experimental results where requested. We are happy to run additional experiments beyond those in our initial responses generally but note that these are compute heavy, slow and costly.

---

### Meta-Review · Area_Chair_xcB1 · 2026-01-03

**Summary:**

Reviewers generally agree that the paper addresses an important and high-stakes problem, and that the proposed FFT method is technically sound. However, significant concerns were raised by Reviewers pjMC and SH1w regarding the breadth of the contribution. In particular, they questioned whether FFT represents a fundamentally new methodological advance or primarily an engineering refinement over existing embedding-based calibration approaches. On the empirical side, multiple reviewers noted that reported improvements often appear numerically small, raising concerns about the decisiveness of the gains. The initial restriction to older backbone architectures further limited perceived generality. While the rebuttal introduced additional experiments, concerns remain about whether these additions sufficiently elevate the methodological impact. Overall, the decision was primarily informed by unresolved doubts about novelty depth, and analytical insight.

**Reviewer Concerns:**

Partially addressed concerns are listed as follows. The authors convincingly addressed concerns about limited backbones by adding experiments on a Vision Transformer (Swin-T).Additional experiments under Gaussian noise and JPEG compression partially addressed robustness concerns.
Outstanding concerns are listed as follows. Despite rebuttal arguments, Reviewers pjMC and SH1w remained unconvinced that FFT goes beyond a well-engineered calibration heuristic. Requests for deeper theoretical grounding and principled analysis remain largely unmet.
Concerns about tightly clustered results and absence of confidence intervals were not fully resolved. While the authors argue this follows community norms and is computationally costly, reviewers emphasized that the marginal nature of improvements makes such analysis particularly important. Furthmore, some reviewers continue to view the gains as modest relative to strong baselines, especially outside hard-shift scenarios, limiting perceived impact.

**Reviewer Scores:**

Reviewer aCaf: Likely to maintain their already positive score. Reviewer pjMC: Unlikely to change their score. Despite added experiments, their core concerns about novelty and impact appear fundamentally unresolved, so the score would likely remain. Reviewer SH1w: Explicitly stated they would keep their score unchanged. While they acknowledged improvements on architecture and ablations, they confirmed that concerns about analytical depth and statistical robustness persist.

---

### Decision · Program_Chairs · 2026-01-26

Reject